# Towards Robotic Pruning: Automated Annotation and Prediction of Branches for Pruning on Trees Reconstructed Using RGB-D Images

**DOI:** 10.3390/s25185648

**Published:** 2025-09-10

**Authors:** Jana Dukić, Petra Pejić, Ivan Vidović, Emmanuel Karlo Nyarko

**Affiliations:** Faculty of Electrical Engineering, Computer Science and Information Technology Osijek, 31000 Osijek, Croatia; ivan.vidovic@ferit.hr (I.V.); karlo.nyarko@ferit.hr (E.K.N.)

**Keywords:** autonomous fruit tree pruning, RGB-D images, 3D tree reconstruction, automatic annotation

## Abstract

This paper presents a comprehensive pipeline for automated prediction of branches to be pruned, integrating 3D reconstruction of fruit trees, automatic branch labeling, and pruning prediction. The workflow begins with capturing multi-view RGB-D images in orchard settings, followed by generating and preprocessing point clouds to reconstruct partial 3D models of pear trees using the TEASER++ algorithm. Differences between pre- and post-pruning models are used to automatically label branches to be pruned, creating a valuable dataset for both reconstruction methods and training machine learning models. A neural network based on PointNet++ is trained to predict branches to be pruned directly on point clouds, with performance evaluated through quantitative metrics and visual inspections. The pipeline demonstrates promising results, enabling real-time prediction suitable for robotic implementation. While some inaccuracies remain, this work lays a solid foundation for future advancements in autonomous orchard management, aiming to improve precision, speed, and practicality of robotic pruning systems.

## 1. Introduction

In this paper, the task of detecting branches to be pruned on fruit trees is addressed. This step is essential for automating the pruning process. Once the branches requiring pruning and the cutting points are accurately identified, a robotic system can execute the pruning accordingly. To make this feasible, the robot must have 3D information about the position and orientation of these pruning points. Therefore, the detection of branches to be pruned is performed on 3D models, specifically point clouds.

Since this represents a real-world problem, an approach that automatically detects branches to be pruned directly on point cloud models of actual trees is proposed. To obtain these point clouds, RGB-D images of real trees were captured directly in orchards, facilitating the potential for the entire process to be performed by a robot in real time in the future. This work serves as a proof-of-concept, demonstrating that the proposed approach is capable of producing results sufficient for practical applications, considering both accuracy and processing speed. Limitations of the current method are acknowledged, providing a foundation for future research and improvements. Nonetheless, the pipeline developed is considered a solid starting point, offering meaningful results for real-world robotic pruning applications (The repository with code and dataset can be found here: https://github.com/ividovic/BRANCH_pipeline, accessed on 7 September 2025).

The initial step involved acquiring RGB-D images of the pear trees. Multiple viewpoints were used during image collection to ensure comprehensive coverage of each tree, which is crucial for accurate 3D reconstruction. Depth information was particularly emphasized due to its importance in robotic applications. Image acquisition took place in the orchard of the Agricultural Institute in Osijek, ensuring that the trees could be pruned by experts. RGB-D images of the same trees were also captured after pruning. Point clouds were generated from these images, followed by preprocessing to remove background elements unrelated to the tree, as well as ground segments. However, the removal of neighboring trees and supporting trellis wires was considered impractical, as manual extraction would have been very time-consuming. Furthermore, due to the sparsity of data, it was extremely difficult to distinguish at the point level which points belonged to the target tree versus adjacent neighboring trees. An approach was developed for the registration of these point clouds to create a reconstruction of the 3D model of each tree, referred to as partial 3D models, as the image acquisition did not form a closed loop. The registration was performed using the TEASER++ algorithm.

For machine learning training in pruning branch detection, ground truth points belonging to the branches to be pruned were automatically labeled. This was achieved by registering the pre- and post-pruning models; the differences between the aligned point clouds were assumed to correspond to pruned branches, which were then labeled accordingly. The neural network training utilized PointNet++, with an ablation study conducted to assess the influence of hyperparameters such as batch size, learning rate and number of points, and also voxel size and normalization of the point clouds. Model evaluation included both quantitative metrics—precision and recall—and qualitative visual inspection.

The main contributions of this work are summarized as follows:

1. BRANCH_v2 dataset: An enhanced version of the BRANCH dataset [1], containing RGB-D images and reconstructed partial 3D models of pear trees pre- and post- pruning. Points belonging to the branches to be pruned are annotated. The dataset containing pre- pruning partial models with corresponding annotations is provided in multiple variations, including normalized and centered forms with voxel sizes ranging from 2.5 mm to 2 cm, totaling ten dataset versions. This dataset can serve as a valuable resource for future research aimed at both tree reconstruction and the prediction of branches to be pruned.

2. Point Cloud Registration Method: Development of a robust point cloud registration method using the TEASER++ algorithm, enhanced with an automatic preprocessing step for background removal. This improves alignment accuracy in sparse, partially overlapping tree structures, and reduces the impact of outliers and environmental noise.

3. Automatic Branch Labeling: A method that automatically labels branches to be pruned, eliminating the necessity of manual annotation.

4. End-to-End Real-Time Pipeline: A complete pipeline that processes RGB-D images to predict pruned branches in real time under real-world conditions. The pipeline’s effectiveness has been validated through neural network training and evaluation, both quantitatively and visually, underlining its potential for future integration into robotic pruning systems.

The paper is structured as follows. Section 2 provides the overview of the research domain and a summary of prior approaches. Section 3 describes the dataset acquisition process, the preprocessing algorithm, and the overall structure of the dataset. In Section 4 the reconstruction of partial 3D tree models is presented. Section 5 explains the annotation of these models and the implementation of a deep neural network for prediction of branches to be pruned. In Section 6 the quantitative and visual evaluation of model predictions in comparison to ground truth labels is presented. Section 7 discusses the main achievements and possibilities for the improvements of the work, while the paper concludes with a summary in Section 8.

## 2. Related Research

The automation of agricultural tasks such as pruning is a rapidly advancing field driven by the need to address labor shortages and enhance efficiency and yield. Recent research heavily leverages machine vision, deep learning, and robotic systems to achieve precision agriculture.

### 2.1. Datasets

High-quality datasets are essential for reliable machine learning models. These can be synthetic data, generated quickly and cheaply, or real data obtained through direct image acquisition, which is often costly and time consuming. However, it seems that there are not enough comprehensive and high quality publicly available datasets with the necessary annotations that are suitable for solving automation problems in orchard pruning. Synthetic datasets, such as SynthTree43k [2], provide large amounts of data, but suffer from a reality gap that affects performance in the real world. Real-world datasets, such as those in [3,4,5,6,7], are limited in size and scope, often manually annotated, and sometimes not publicly available. For example, in [3], nine apple trees were captured using a Kinect2 sensor, while [6] provides a comprehensive dataset with 11,000 labeled vineyard images. Additionally, ForestSemantic was introduced in [7] for forest segmentation. Our previous work [1] proposed BRANCH, a pear tree RGB-D dataset with reconstructed 3D models. In this paper, BRANCH dataset is extended with more trees, improved reconstructed 3D models, and automatic pruning annotations.

The previously mentioned studies are summarized and compared with the BRANCH_v2 dataset, acquired during the research proposed in this paper, in Table 1.

### 2.2. 3D Tree Reconstruction

The method proposed in [8] is based on reconstructing a 3D model of an apple tree using the shape-from-silhouette technique, where images are captured from different angles using a Time-of-Flight (ToF) 3D camera. All images are initially processed to remove noise and unwanted gaps. The tree skeleton is then built using a medial axis thinning algorithm. These models are analyzed to detect the trunk and identify branches, achieving 100% accuracy in trunk detection, while branches are detected with 77% accuracy. Lengths of branches, gaps between them, and overlapping branches are also estimated, which are emphasized as critical information for automatic pruning. However, the main drawback of this method is the high computational requirements, which prevent real-time application. The authors suggest improving the method by preprocessing 3D data into 2D to obtain a tree skeleton, then converting it back into 3D space—something planned for future testing.

In [9], a generative statistical approach for the extraction of 3D branch structure extraction based on image sequences is proposed, where trees are modeled in 3D using L-systems. Due to poor contrast, background noise, or variable branch projection order, a Markov Chain Monte Carlo (MCMC) method is used for extraction, which utilizes mutual correlation and allows the generation of complex and uncertain branch models. This work demonstrates that 3D models can only be generated with consumer-grade cameras when the tree is recorded in wide-angle sequences. However, more advanced sensors are required for practical applications [10].

Building on the work of [9], authors of [11] employ a similar approach using data captured with two laser devices. This allows modeling complex sensor systems without solving complicated inverse problems, although challenges remain due to the complexity of sensors and noise. The goal was to develop a robust method that preserves depth information, represented as cylinders, with noted limitations in system complexity and initial errors.

For modeling a tree from point clouds from multiple images, common methods include registration techniques such as the iterative closest point (ICP) algorithm or Chen and Medioni’s method, which perform well with good initial alignment [12]. However, for partial and unorganized point clouds, these methods are less suitable due to difficulty in matching corresponding points [13]. Authors in [6] propose a method based on geometric primitives and neighborhood search, utilizing changes in curvature and normal vectors to match clouds, with testing on simulated and real scan data.

Finally, the feasibility of reconstructing 3D tree models using RGB-D cameras was investigated in [1,14]. In [14] it was shown that 3D models can be reconstructed from only two perspectives; however, outdoors this is limited by lighting conditions. These issues can be mitigated by using higher-quality sensors designed for outdoor use or recording during low-light conditions (early morning or late evening). The method involves merging two RGB-D images captured from different positions within three meters of the tree using known transformation matrices. Errors in these matrices lead to inaccurate models. A similar conclusion is drawn in [1].

Frequently, captured point clouds in real time often contain added noise which cannot be completely removed. These noise points pose a challenge for some of the previously described registration methods. Therefore, authors of [15] present the TEASER algorithm, enabling fast and reliable registration of point clouds in 3D space even in the presence of errors and noise. The algorithm employs a rigid transformation approach, which includes translation, rotation, and scaling to accurately align the given point clouds. Its key advantage is the mathematical proof of the solution’s accuracy. TEASER is applied in the reconstruction process proposed in this paper.

### 2.3. Automation of Pruning

In [16], the authors introduce an improved YOLOv5-based tomato pruning point segmentation model (TPS-YOLO), which demonstrates greater robustness in segmenting stems, lateral, and fruiting branches in complex environments. Using the segmentation output, they developed a pruning point localization algorithm that filters candidate points and identifies the optimal pruning point with an evaluation box mechanism. This method achieves a precision of 92.0% and recall of 88.9%, enabling real-time localization of pruning points and supporting automated pruning.

A study on automatic pruning of cherry tree canopies is presented in [17], where pruning decisions are based on the Upright Fruiting Offshoots (UFO) architecture. The study employs an instance segmentation approach using Mask R-CNN to detect “leaders” (vigorous upright shoots) and estimate their diameters—key factors in UFO pruning—to identify the largest leader for removal. The authors found that models trained with active lighting images produced more robust pruning decisions compared to those trained with ambient light images.

Similar to cherry pruning, winter pruning of grapevines is labor-intensive and repetitive, requiring skilled workers. Authors of [18] propose an autonomous approach that combines image processing and artificial intelligence. Building on prior 2D segmentation of grapevine organs, the algorithm is extended to 3D using point clouds generated from depth images and segmentation masks. The system extracts thickness measurements and applies agronomic rules to determine pruning points for balanced pruning. Field trials in a potted vineyard achieved an accuracy of 54.2%. A key feature is the customizable framework, allowing users to adjust parameters such as the number of nodes retained on spurs and cane thickness, demonstrating the versatility of their 3D system.

A comprehensive review by the authors in [19] provides an overview of technological advances in apple production and covers six different areas: Pruning, pollination, thinning, bagging, harvesting, and sorting. The section on apple pruning discusses the application of vision systems and mechanical branch cutting methods in automated apple pruning. Pruning is highlighted as a cumulative process that is critical to plant growth, productivity and yield optimization. It is emphasized that automated pruning involves the recognition of tree geometry, the implementation of pruning procedures and the use of automatic navigation for branch pruning. The report also looks at how Artificial Intelligence (AI) technologies have become prevalent across various areas of apple production, benefiting the economy and increasing productivity. This comprehensive report summarizes existing research, identifies challenges and outlines future prospects for increasing the quality and production of apple fruit through automation.

In [20], a vision-based algorithm for pruning dormant cherry trees is presented. This approach is noted as the first in the literature to propose a vision-based precision pruning algorithm based on strict pruning rules that determine exact pruning points for dormant cherry trees and can be easily adapted to different tree species by adjusting the pruning rules. The method includes a semantic multi-class segmentation using a U-network with a VGG16 backbone to recognize trunks, branches and shoots. Geometric calculations based on specific pruning strategies are then used to precisely locate the pruning points. The method for determining the cutting points resulted in an average accuracy of 93.33%, with precision rates of 88.75% for branches, 91.25% for shoots and 100% for trunks.

In comparison, our work differs significantly: while [20] models specific types of pruning based on predefined rules, we aim to emulate how a human or an expert performs pruning, focusing on learning from data rather than explicitly defining rules. Additionally, their dataset was manually annotated for training purposes, while we aim to automate this annotation process. While their approach relies on rule-based geometric calculations to identify pruning points, our goal is to leverage deep learning techniques to predict pruning points directly from 3D data, allowing for greater flexibility and adaptability to different pruning scenarios.

## 3. Dataset

Recognizing the limitations of existing data in agricultural robotics, an extended version of the BRANCH dataset, BRANCH_v2 (The complete dataset is available here: https://puh.srce.hr/s/EoPqgASGerLapne (accessed on 7 September 2025), or can be obtained upon request from the authors), was created in collaboration with the Agricultural Institute in Osijek, Croatia. In our previous work [1], the original BRANCH dataset was introduced, which comprised RGB-D images of 70 pear trees which resulted in 52 reconstructed tree models, captured both before and after pruning, with labeled points belonging to the branches for pruning. In this study, we build upon that dataset by incorporating an additional RGB-D images of 114 trees, covering one full row of 184 pear trees in the orchard, thereby significantly expanding the dataset’s scope and potential for analysis. The RGB-D images were captured using an ASUS Xtion PRO Live RGB-D camera, with a resolution of 640 × 480 pixels, which was sufficient for generating point clouds and performing 3D reconstruction. Image acquisition took place in late winter, when the trees were in a dormant state—characterized by the absence of leaves, with only bare branches and trunks visible. To ensure high-quality data, the dataset was collected at sunset, exclusively under conditions without rain or wind. The camera was positioned at approximately 10 viewpoints along the front side of each tree, as the presence of trellis wires prevented full 360-degree scanning, enabling creation of the partial tree 3D model. The data presented in this paper consists of three sets. The first set includes RGB-D images of pear trees taken before pruning, along with a point cloud for each view and a single partial 3D model reconstructed for each tree. The second set of the dataset contains RGB-D images captured after an expert from the Agricultural Institute pruned the entire row; accompanied by corresponding point clouds for each view and partial 3D model for each tree. In this paper, the term model specifically refers to a point cloud representation of a tree. Finally, the third set includes registered partial models from pre- and post- pruning, with thus obtained labeled branches identified for pruning. By overlapping the pre- and post-pruning models, data-driven ground truth data were generated. This approach is more practical and cost-effective than manually annotating every point in the point cloud. It enables experts to perform pruning in the field, with the proposed method automatically labeling the relevant points, eliminating the need for time-consuming point-by-point annotation. The pruning technique applied in this orchard aimed to improve fruit quality by removing excess branches that primarily consumed water without contributing to fruit production. Note that the post-pruning images were taken once the trees had already begun to blossom.

Each tree was partially reconstructed using only four RGB-D images, carefully selected to provide sufficient overlap of the point clouds and to reconstruct almost the entire tree. An example of approximate selection is shown in the Figure 1 illustrating camera positions at mid-left (colored red), center (green), mid-right (blue), and center-upper (yellow) viewpoints. The choice of using exactly four images per tree was driven by computational constraints, as the registration process is resource-intensive, and including more images did not significantly improve the final model quality. The novelty of this work, compared to [1], lies in the enhanced preprocessing approach, which includes the removal of grass and the application of the TEASER++ algorithm [15] to improve the robustness and accuracy of point cloud registration. Overall, the dataset comprises RGB-D image captures of 184 individual pear trees, providing a substantial basis for 3D reconstruction and pruning analysis. Each reconstructed 3D partial model was visually examined. If the trunk and at least two primary branch points overlapped with minimal disparity, the registration was deemed successful. On the other hand, even a slight tilt of the trunk or lack of overlap at the branch points was considered a failed registration. Using the described approach, a total of 101 partial 3D tree models both before and after pruning process were proclaimed successfully reconstructed, with labeled points indicating the pruned branches.

### 3.1. Dataset Structure

The dataset is organized into three main folders: A, B, and Merged, corresponding to the images taken *after (A)* pruning, *before (B)* pruning, and the *aligned* (merged) data, respectively. Within each of these folders, a subfolder is named with the prefix E, which stands for evening capture session. Each E folder contains data for several trees, which are structured as follows:


tree_<row>_<cultivar>_<iteration>


Here, <row> corresponds to the tree row in orchard, <cultivar> corresponds to the Williams pear cultivar (shortened ‘V’ in Croatian) and <iteration> indicates the specific tree order number. Inside each tree folder, there are three subfolders:original_images/ – containing unprocessed datafiltered_noGrass/ – containing preprocessed data with removed grassreconstruction/ – containing registered data

Both original_images/ and filtered_noGrass/ folders include:RGB images (.png)Depth images (.png)Corresponding point clouds (.ply)

The reconstruction folder contains the (.ply) registered partial 3D model with voxelized point clouds used for the registration process.

The Merged folder contains final overlap of the pre- and post- pruning models and additionally labeled branches for pruning on point-level annotation provided as a csv file. This file is organized as follows: voxel size, tree name, and labels. The labels correspond to the registered point cloud and are represented as a list of ones and zeros, where a value of 1 indicates that the point belongs to a pruned branch, and a value of 0 indicates non pruning point. This hierarchical structure facilitates organized access to raw and pre-processed data for training and evaluation purposes. Point clouds derived from the RGB-D images before the pruning were further processed for training, including centering, normalization, and voxelization at eight different voxel sizes ranging from 2.5 mm to 2 cm. The labels were adjusted accordingly, resulting in a total of 10 dataset variations.

### 3.2. Preprocessing Dataset

The process developed in this research is intended to be applicable in real-world conditions. Accordingly, the RGB-D images were collected directly in an outdoor orchard setting, capturing all environmental factors a robot would encounter during operation. The entire dataset was uniformly preprocessed and then randomly split into training and testing subsets, ensuring similar environmental conditions in both. The dataset already exhibits significant occlusion from the tree’s own branches and trunk, as well as from neighboring trees and structures like wires, present in both subsets. Since the approach relies solely on depth information, additional augmentation for lighting perturbation would offer limited benefit. Therefore, only grass removal preprocessing was implemented, as described below.

Each point cloud was generated using the corresponding color and depth images along with the camera’s intrinsic parameters, as described in [1]. Due to the specific imaging setup, all point clouds shared a similar spatial layout: a tree was always positioned in the foreground—typically centered, though occasionally with overlapping branches from neighboring trees. The background sometimes contained distant trees or grass. The bottom part of the scene often included grass; however, in some cases, it was not visible because the camera was elevated to capture the upper portions of the tree. This observation motivated the development of an automated grass removal algorithm, presented in Algorithm 1. The preprocessing pipeline is demonstrated using the before pruning image seen in Figure 2a. The input to the algorithm is a point cloud *P*.

Since the camera was positioned vertically to capture only one tree per frame, the resulting point cloud required reorientation to achieve an upright tree alignment. Accordingly, the initial preprocessing step involved rotating *P* about the Z and Y axes of the camera’s reference frame to ensure proper vertical alignment of the tree.
**Algorithm 1:** GrassRemoval (P): Grass removal from point cloud *P*  1:**procedure** GrassRemoval(*P*)  2:       Rotate *P* about Z and Y axes to align vertically.  3:       **if** $\;|P_{z,max}-P_{z,min}|>1.5$ m **then**  4:             Flag *P* as Potential_Grass  5:             $B\leftarrow \{p\in P|p_{y}<P_{y,min}+0.6\;m\}$  6:             Planes←∅  7:             **while** |B|≥30 **do**  8:                  (π,Inliers)← RANSAC_Plane_Segment(*B*)  9:                  **if** |π|<30 **then**10:                       **break**11:                  **end if**12:                  θ← angle between normal of π and *Y*-axis13:                  **if** $\theta \leq 35^{\circ }$ **then**14:                       Planes←Plane∪{π}15:                  **end if**16:                  B←B∖Inliers17:             **end while**18:             **if** Planes≠∅ **then**19:                  $\pi_{\mathrm{main}}\leftarrow$ plane in Planes with maximum number of points20:                  $B\leftarrow \{p\in P|p_{y}<P_{y,min}+0.7\;m\}$21:                  G←∅22:                  **for all** p∈*B* **do**23:                       d← distance from *p* to $\pi_{\mathrm{main}}$24:                       **if** d≤0.35 m **then**25:                            G←G∪{p}26:                       **end if**27:                  **end for**28:                  $P_{clean}\leftarrow P\setminus G$29:                  **return** $P_{\mathrm{clean}}$30:             **else**31:                  $B\leftarrow \{p\in P|p_{y}<P_{y,min}+0.6\;m\}$32:                  $T\leftarrow \{p\in P|p_{y}>P_{y,min}+0.6\;m\}$33:                  $z_{\min}^{T}\leftarrow \min\left\{p_{z}|p\in T\right\}$34:                  $G\leftarrow \{p\in B|p_{z}<z_{\min}^{T}\}$35:                  $P_{\mathrm{clean}}\leftarrow P\setminus G$36:                  **return** $P_{\mathrm{clean}}$37:             **end if**38:       **else**39:             **return** *P*               ▹ No preprocessing needed40:       **end if**41:**end procedure**

The next preprocessing step involved removing ground-level grass and distant background points from the point clouds. In previous work [1], grass removal was performed using the K-Means clustering algorithm with two clusters to separate grass from the tree. However, due to limited computational resources at the time, further preprocessing steps were not feasible, and the selection of images for grass removal was done manually. In the current work, this process was significantly improved and automated to enhance consistency and efficiency.

To automate detection and removal of points belonging to the grass, a heuristic method based on spatial thickness was implemented. Based on empirical data, measured in orchard, each tree was expected to occupy approximately 1.5 m in width along both the X- and Z-axes. Specifically, if the absolute distance of *Z*-coordinates in the considered point cloud *P*, hereafter denoted as $\Delta P_{z}=P_{z,max}-P_{z,min}$, is greater than 1.5 m, the point cloud was flagged as potentially containing grass or objects in the background. Otherwise, no further preprocessing was applied.

Given that the average grass height was empirically measured to be approximately 0.4 m during dataset acquisition, a threshold of $P_{y,min}$ + 0.6 m along the Y-axis was applied. All points below this height were considered potential grass. This additional 0.2 m provided a margin for potential errors caused by camera tilting, ensuring all grass points could be detected. Although a threshold of 0.2 m may seem large, it primarily affected the lower portion of the trunk, which was not critical for our purposes. Even if this section was mistakenly cropped, it would have minimal impact on the resulting registration. Consequently, a new point cloud containing only the bottom part of the flagged point cloud *P*, up to the height of $P_{y,min}$ + 0.6 m along the Y-axis—is defined as *B* to ensure faster grass detection, as illustrated in Figure 3a. Points further than 1.5 m in the negative Z direction were presumed to lie outside the target tree structure and were removed accordingly.

For the point cloud *B*, iterative RANSAC-based plane segmentation was performed using the segment_plane() [21] method from the Open3D library. In each iteration, the detected plane inliers were removed from the remaining point cloud *B*. This removal step was necessary because the segment_plane() function tends to repeatedly detect the same planes if previously segmented points are not excluded. Furthermore, let $\vec{n}$ be the unit normal of the plane and $\vec{y}=\left(0,1,0\right)$ represent the vertical axis. The angle θ between $\vec{n}$ and $\vec{y}$ was computed as:$$\theta =\arccos(\vec{n}\cdot \vec{y})$$A detected plane was retained only if it met two criteria:1.Minimum number of inliers: N≥30,2.Plane orientation: $\theta \leq 35^{\circ }$

If $\theta \leq 35^{\circ }$, the plane was considered approximately horizontal and retained. The $35^{\circ }$ angle threshold was empirically determined based on the camera’s tilt during data acquisition.

The segmentation process continued until fewer than 30 points remained in the bottom point cloud *B*. A visualization of the segmented planes representing grass—highlighted in red and purple—can be seen in Figure 3b. In the next step, only the dominant plane $\pi_{main}$ containing the highest number of inlier points, was selected, as this plane consistently corresponded to a horizontally aligned surface resembling the characteristic pattern of grass (e.g., the red plane in Figure 3c). Other segmented planes, which had fewer amount of points and larger tilt angles, were often found intersecting lower parts of the tree structure (e.g., trunk), mistakenly including tree points as part of the grass. By focusing on the most dominant horizontal plane, this approach minimized the risk of removing valid tree geometry and improved the accuracy of the grass segmentation process. Additionally, it was necessary to incorporate points from other segmented planes that also represented the grass surface, visible as black points near the red plane in Figure 3c. To achieve this, the point cloud *B* was updated by selecting points from the original point cloud *P* that satisfied the condition of having a Y-axis value less than $P_{y,min}$ + 0.7 m. This elevated threshold accounts for potential inclinations in the grass plane. For each point p=(x,y,z) in *B*, its orthogonal distance *D* to the plane was calculated as follows:$$D=\frac{\left|ax+by+cz+d\right|}{\sqrt{a^{2}+b^{2}+c^{2}}}$$
where (a,b,c) are the plane normal coefficients and *d* is the plane offset ($P_{y,min}$ + 0.7 m). If D≤0.35m, the point was considered part of the dominant grass plane.

Finally, all points within ±0.35 m of the dominant plane were labeled as grass and removed from the original point cloud *P*, while the remaining points were retained as part of the tree structure.

In cases where RANSAC failed to detect any or valid planes, grass points were heuristically identified based on their spatial location. Specifically, the points from the original point cloud *P* were divided into two subsets: *B*, representing the lower region of the scene, defined as points with $P_{y}<P_{y,min}+0.6$ m; and *T*, representing the upper part of the point cloud, with $P_{y}>P_{y,min}+0.6$ m. The minimum Z-value among the points in *T* was computed and denoted as $z_{\min}^{T}$. Then, a subset of points G⊂B was defined as those points in the lower region whose Z-coordinate was less than this threshold: $z_{\min}^{T}$, i.e., $G=\{p\in B|P_{z}\left(p\right)<z_{\min}^{T}\}$. These points in *G* were identified as likely belonging to distant background structures (e.g., grass or terrain behind the tree) and were removed from the original point cloud. The resulting cleaned point cloud was obtained as $P_{\mathrm{clean}}=P\setminus G$.

The output of the algorithm is the point cloud of the tree without grass or background structures, denoted as $P_{clean}$, in cases where grass was initially detected or objects were present in the background (see Figure 3d).

## 4. Tree Reconstruction

Having clear, noise-free point clouds is important when applying reconstruction algorithms. Many commonly used methods- such as Iterative Closest Point (ICP)—tend to perform poorly in the presence of noise and outliers [22]; while RANSAC’s performance degrades significantly as noise level increases [23]. An additional challenge arises when reconstructing unordered and sparse objects, such as trees, from multiple viewpoints: the resulting point clouds often have minimal overlap, making it difficult or even impossible to establish reliable correspondences purely through mathematical means, especially in the absence of perceptual context.

The reconstruction process closely follows the methodology described in our previous work [1]. For each tree, four point clouds, subjectively selected based on visual assessment as containing the most relevant structural information, were manually chosen for the reconstruction algorithm. Due to computational constraints, using more than four point clouds would have made registration overly complex; therefore, the selected four were carefully chosen to avoid redundant views and to exclude those lacking meaningful structural information (e.g., point cloud showing only a single branch). In cases where more than four high-quality point clouds were available, preference was always given to those capturing a greater number of additional branches. Figure 4 illustrates the importance of manual point cloud selection. Successful registration using four manually chosen point clouds is shown in Figure 4a. In contrast, Figure 4b demonstrates that using fewer than four point clouds leads to incomplete reconstruction. Figure 4c shows a failed registration when four point clouds are selected at random, while Figure 4d depicts registration failure when all six available point clouds are used. These results highlight that careful manual selection of representative views is crucial for reliable reconstruction. Furthermore, by systematically observing which viewpoints consistently lead to successful registrations, it will be possible in future work to predefine optimal acquisition positions, thereby eliminating the need for manual selection and enabling a fully automated pipeline. As detailed in Section 3.2, point clouds were first preprocessed; and then sequentially registered. In our previous work [1], the registration process employed global registration (RANSAC) as an initial step for the local ICP method. The ICP algorithm has several limitation: accuracy of the results relies on a good initial alignment and can be affected by point cloud imperfections such as noise, outliers and partial-overlaps [24]. In this paper the object of interest are tree structures, which are sparse and not entirely rigid due to the potential wind-induced moving. These characteristics, combined with small overlapping areas in chosen images and noise introduced by neighboring branches, pose significant challenge for standard ICP registration. In such conditions characterized by limited overlap, structural sparsity, and nonrigid deformations, ICP is likely to converge to incorrect alignments or fail altogether. A review of the literature highlights fast and certifiable algorithm called Truncated least squares estimation And SEmidefinite Relaxation (TEASER++) for the registration of two sets of point clouds with a large number of outlier correspondences [15]. TEASER++ is designed to provide certifiable global optimality in the presence of extreme outlier rates- handling up to 99% of outliers - by formulating the rotation estimation as a semidefinite program (SDP) and solving it efficiently through convex relation techniques. Unlike traditional methods such as ICP, TEASER++ is not only more accurate, but also significantly faster, often completing tasks in milliseconds [15]. Its certifiability, robustness to outliers, and deterministic success bounds make it especially suitable for registering sparse, partially overlapping structures such as trees.

The algorithm for pear tree reconstruction is presented as Algorithm 2, which is explained using a pre-pruning dataset example seen in Figure 5. The input consists of four preprocessed point clouds: $P_{0},P_{1},P_{2},P_{3}$, representing different views of the tree. When combined, these provide almost complete scan of the tree. The first point cloud is always designated as the stationary reference and is referred to as the target point cloud, $P_{0}$ in Figure 5a, while the remaining point clouds are considered source point clouds, in Figure 5b–d. The algorithm is designed in a way that the final integrated point cloud remains aligned within the reference frame of the initial target point cloud.
**Algorithm 2:** RegistrationTEASER $\left(\left\{P_{0},P_{1},P_{2},P_{3}\right\}\right)$: Sequential TEASER++ registration  1:**procedure** RegistrationTEASER($\left\{P_{0},P_{1},P_{2},P_{3}\right\}$)  2:      $target\leftarrow P_{0}$  3:      **for** k=1 to 3 **do**  4:            $source\leftarrow P_{k}$  5:            source_,target_←Downsample(source,target)  6:            FPFH_s,FPFH_t←FPFH_features(source_,target_)  7:            source_corr,target_corr←Correspondences(FPFH_s,FPFH_t)  8:            $transformation_{TEASER}\leftarrow TEASER\_registration\left(FPFH\_s,FPFH\_t\right)$  9:            $source_{T}\leftarrow Transform\left(source,transformation_{TEASER}\right)$10:            $target+=source_{T}$11:      **end for**12:      **return** target13:**end procedure**

Since each preprocessed point cloud contains more than 40,000 points, voxel downsampling was performed for both the source and target clouds to improve computational efficiency. A voxel size of 0.02 m was chosen as an effective compromise between registration accuracy and runtime performance. Preliminary tests showed that smaller voxel sizes (e.g., 0.01 m) preserved fine geometric details but significantly increased computation time and occasionally led to unstable registrations. Conversely, larger voxel sizes (e.g., 0.03 m) reduced computational costs, but resulted in insufficient point density, and thus lower alignment accuracy. The visual evaluation of the registration results in combination with these computational considerations, indicated that a voxel size of 0.02 m provided the best balance between accuracy and efficiency. TEASER++ relies on Fast Point Feature Histogram (FPFH) [25] descriptors to establish correspondences between point clouds [15]. First, FPFH features are extracted from the downsampled source and target point clouds (source_ and target_). Using KDTreeFlann [21], nearest neighbor searches are conducted on the target features for each source feature, producing arrays of source and target correspondences. Next, global registration is performed using TEASER++ with the identified correspondences (*source_corr*, *target_corr*). The open-source TEASER++ library [26] (C++ with Python and MATLAB bindings) is used for this step. The TEASER++ parameters employed for the registration process are summarized in Table 2. Among them, two parameters were adjusted from their default values: the noise_bound was set to 0.02, matching the voxel size used for downsampling, instead of the default value of 1, and the rotation_estimation_algorithm was set to False, since additional rotation estimation was unnecessary. The output is a transformation matrix that maps the source point cloud into the target point cloud’s reference frame. In this paper, the target point cloud is consistently colored blue, and the source point cloud red. The resulting registration after applying computed transformation matrix on source point cloud, shown in Figure 6a, demonstrates good overlap in key areas such as the tree trunk, a prominent vertical branch, and the horizontal support wire near the base of the scene. In the next step, the target point cloud is updated to include the previously registered source point cloud, with the combined target point cloud. In next iteration, the new source point cloud, shown in Figure 5c, is registered against this updated target after finding correspondences and suitable transformation, resulting in the point cloud seen in Figure 6b. This iterative process continues until all point clouds have been registered. The final result of combining all four point clouds, shown in their original colors, is presented in Figure 6d. It is important to note that while the transformation matrices are computed using the downsampled point clouds, these matrices are ultimately applied to the original, full-resolution source point clouds for final storage and visualization.

The same process was also applied to generate models from the after pruning images. An example of a resulting post-pruning model is shown in Figure 7b. Comparing the pre- pruning model (Figure 7a) with the post- pruning model (Figure 7b) clearly illustrates the extent of branch removal due to pruning.

A limitation of this sequential point cloud registration approach is the accumulation of errors. If an error occurs early in the sequence, it can propagate and negatively affect all subsequent transformations. However, since the images were captured in a strict, consecutive order, it is reasonable to assume that each newly added point cloud shares increasing overlap with the existing model, facilitating easier registration. An alternative approach, registering only neighboring point clouds, was also tested but frequently failed when consecutive images lacked sufficient overlap, leading to incorrect registration. An additional limitation of this work is the reliance solely on visual inspection for evaluating each step of the registration. In previous work [1], Intersection over Union and Chamfer Distance were employed, but in this study, these metrics—together with RMSE—proved unreliable for detecting incorrect registrations in the absence of ground truth. The challenge of robust evaluation warrants further investigation, and the development and adoption of additional metrics will be addressed in future work.

Another improvement over the previous work was the substitution of the ICP algorithm with TEASER++, which required only a single run per registration. In contrast, the earlier approach necessitated running ICP five times for each registration to increase the likelihood of successful alignment. This repetition significantly increased computation time, as ICP often failed to achieve satisfactory alignment on the initial attempt. Moreover, if a suitable overlap was not found within these five iterations, the registration typically failed entirely.

As mentioned in Section 3, images were captured for a total of 184 trees, representing a complete row in the orchard. These trees were captured both before and after the pruning process. All generated point clouds were preprocessed by applying rotation corrections and removing excess grass and background elements. Subsequently, the point clouds were registered using the TEASER++ algorithm for both pre- and post-pruning scans. This approach facilitates the efficient and cost-effective generation of ground truth data, eliminating the need for experts to manually label branches to be pruned, which will be explained in the Section 5.

The TEASER++ registration successfully aligned all four point clouds for 133 trees pre- pruning (72.3%) and 143 trees post- pruning (77.7%), as summarized in Table 3. Among these, 103 trees (55.9%) had both pre- and post-pruning models correctly reconstructed. For these, a final registration was performed using TEASER++ to generate the overlapped pre- and post-pruning models, treating the pre-pruning model as the target (blue in Figure 7c) and the post-pruning model as the source point cloud (red in Figure 7c). After visual inspection, 101 (98.1%) of these combined models were proclaimed as successfully created models.

In Table 4, we provide a timing overview of the reconstruction, including preprocessing and registration phases. The preprocessing step includes grass removal, applied to four selected RGB-D images. Registration involves point cloud voxelization, FPFH feature extraction, correspondence finding, and matching with TEASER++. Since the registration process is performed sequentially, the table reports the elapsed times for each phase: registration of $P_{0}$ and $P_{1}$, then $P_{0,1}$ and $P_{2}$, and finally registration of $P_{0,1,2}$ and $P_{3}$. The results indicate that registration, on average, takes only a few seconds per single tree, while preprocessing generally requires a couple of minutes on average per single tree, making it the main bottleneck of the overall process. Grass removal and registration experiments were conducted on a 64-bit Linux system equipped with an Intel(R) Core(TM) i7-7700 CPU at 3.60 GHz (4 cores, 8 threads, 8 MiB L3 cache) and an NVIDIA GeForce GTX 1060 6GB GPU.

## 5. Prediction of the Branches to Be Pruned

This section provides an overview of the methodology used to predict which branches of a tree should be pruned. It consists of two main parts: first, the annotation of points associated with pruned branches, which involves identifying and labeling points in the 3D models that correspond to removed structures; and second, the implementation of a neural network-based prediction approach using PointNet++, designed to automatically segment pruned branches in complex tree point clouds.

### 5.1. Annotation of the Points Associated with the Branches to Be Pruned

After the partial 3D model reconstruction and registering the pre- and post-pruning tree models, identifying and labeling the pruned branches may be performed. The goal was to highlight regions in the pre-pruning model that correspond to parts of the tree removed during pruning—i.e., areas present in the pre- model but absent in the post-pruning model. This difference is visually illustrated in Figure 7d, where labeled regions for pruning are highlighted in yellow.

To achieve this, a nearest-neighbor distance-based approach was applied. Let $P_{b}$ denote the point cloud of the pre-pruning tree model, and $P_{a}$ the point cloud of the post- pruning model. For each point $p_{i}\in P_{b}$, the Euclidean distance to its closest neighbor in $P_{a}$ was computed using the Open3D compute_point_cloud_distance() [21] function. Formally, the shortest distance is defined as: $d_{i}=\min_{q_{j}\in P_{a}}{∥p_{i}-q_{j}∥}_{2}$. A distance threshold τ=0.03 m was empirically chosen based on typical registration noise and pruning geometry. Points from $P_{b}$ for which $d_{i}>\tau$ were considered to belong to branches that were pruned. These points were subsequently colored yellow in the final visualization to denote removed structures. This procedure was repeated for all points in $P_{b}$, resulting in an automatically labeled ground truth map of pruned branches. In Table 5, the duration of registration between pre- and post-pruning tree models, as well as annotation time are reported. Registration includes point cloud voxelization, FPFH feature extraction, correspondence finding, and matching with TEASER++. In the table, annotation time implies the above explained process of annotation of the points. On average, the whole process takes around 4 s.

A limitation of this annotation method is its sensitivity to occlusions and noise. Specifically, points in the pre-pruning model that are missing from the post-pruning model due to sensor noise, occlusions, or incomplete reconstruction may be incorrectly classified as pruned branches. Additionally, the method can mistakenly label parts of neighboring trees or background clutter as pruned structures if they were present in the pre-pruning model but absent in the post-pruning model.

This labeling method offers several advantages. It enables semi-automated ground truth generation, significantly reducing the need for manual annotation. The approach is objective and repeatable, relying on geometric differences between point clouds rather than subjective human judgment. Additionally, it is scalable to large datasets and cost-effective, making it well-suited for high-throughput pruning analysis in orchard environments. Ideally, when model reconstructions are accurate, and removal of tree-unrelated structures is successful, this method can precisely highlight removed branches without requiring semantic understanding or expert input.

### 5.2. PointNet++ Implementation

For the task of detecting points belonging to the branches to be pruned on fruit trees we used PointNet++ network [27]. PointNet++ is a hierarchical neural network designed for processing point cloud data, which captures local geometric features at multiple scales by recursively applying PointNet network [28]. This network effectively learns both global and fine-grained local patterns in 3D point clouds, making it particularly suitable for our task of segmenting pruned branches in complex tree structures.

The main hyperparameters used during PointNet++ training are summarized in Table 6. For selected parameters, multiple values were tested during ablation studies to investigate their impact on the model’s performance. Specifically, the batch size was chosen from [2,4], and the optimizer was Adam with a weight decay of $10^{-4}$. Two different learning rates, [0.0025,0.005], were evaluated to analyze their influence on convergence and accuracy. The learning rate was decayed by a factor of 0.5 every 20 epochs throughout a total of 100 epochs. The model was trained for 2 output classes: ‘1’ denoting pruned and ‘0’ not pruned branch points.

Although the number of points varies between different point clouds in the dataset, within a single training the number of points per point cloud (npoints) was fixed for all samples. To examine the effect of input size on model performance, we counted the number of points in each tree model within the dataset. The smallest number of points among all models was used as the minimum input size, and the largest number of points was used as the maximum input size for training. For each tree model used for the training and evaluation of PointNet++, the points were randomly chosen. If the tree model contains fewer points than the parameter npoints, some points were chosen multiple times. On the other hand, if the tree model contains more points than the parameter npoints, some of the points were omitted. The selection of random points is performed by numpy.random.choice function with the parameter replace set to True. The dataset was dynamically split into training, validation, and test sets using a 0.7/0.1/0.2 ratio, controlled by a fixed random seed to ensure reproducibility. The split is performed at the tree instance level, meaning that for the input of PointNet++ we are using only annotated pre-pruning tree models. Out of 101 models, 70 models were used for training, 10 for validation and 21 for test.

## 6. Experimental Evaluation

To evaluate the overall process of predicting branches to be pruned, the input for the training of the PointNet++ model consisted of point clouds of partial tree models, along with labeled points obtained by overlapping the reconstructed pre- and post-pruning models. Following training, the model was tested on a separate set, where it predicted pruning points expected to belong to branches to be pruned. Quantitative evaluation was performed by calculating precision, recall, and F1 score, based on comparisons between the model’s predictions and the ground truth labels. To gain a better understanding of the prediction quality, visual inspections were also conducted, allowing for qualitative comparison between predicted points and ground truth annotations. A small ablation study was conducted to assess the impact of various parameters. Since the original point clouds were referenced to the camera capturing the first RGB-D image of each tree—which was inconsistent across the dataset (sometimes captured from the left, other times from the right)—the point clouds were centered to have their origin at the center of all tree points. Additionally, normalization of the point clouds was applied. To accelerate training and evaluate the network’s robustness to various voxel sizes, the normalized and centered point clouds were additionally voxelized into eight different sizes, ranging from 2.5 mm to 2 cm, resulting in a total of ten dataset variations with corresponding ground truth labels. Details regarding these datasets are provided in Table 7, where dataset names indicate voxel size in meters. Examples of the 3D model with different voxel sizes are shown in Figure 8.

Hyper-parameter tuning was also performed, including variations in batch size, learning rate, and the number of points sampled, as shown in Table 6. Training was carried out on a NVIDIA GeForce RTX 4070 Ti, CUDA Version 12.6, 12 GB GDDR6X, which limited the convergence of certain dataset and parameter variations. To assess the stability of the results, experiments were repeated with different random seeds, [10, 42], used for splitting the dataset into training, validation, and test sets. The results showed consistent trends, indicating that the performance of the model was neither significantly affected by the values of the hyperparameters nor by the choice of seed. The training details and test results, including precision, recall, and F1 scores, are summarized in Table 8.

In this application, high precision is particularly crucial because it indicates that the network is not predicting false positives—meaning it is unlikely to mistakenly identify parts of the tree, such as branches or other structures, as candidates for pruning when they should not be. This is important because unnecessary cutting of healthy parts can cause damage to the tree and compromise its overall health and stability. Therefore, ensuring high precision helps minimize the risk of incorrect pruning and preserves the integrity of the tree.

While quantitative metrics such as precision and recall provide valuable indicators of the model’s performance, they alone are insufficient to fully understand the causes of false predictions. These metrics do not offer insight into the specific reasons behind incorrect classifications, such as whether false positives are due to ambiguous features or ground truth inaccuracies. To gain a deeper understanding of the model’s behavior and to better assess its practical applicability, a visual inspection of the test samples was conducted. This qualitative analysis allows us to observe the pattern, context, and potential sources of errors that are not apparent from numerical metrics alone. We conducted a qualitative validation of the test samples from the chosen training variation with batch size = 2, learning rate = 0.0025, seed = 42, and npoints = min_points. Specifically, we selected the dataset with voxelization at 2.5 mm, as this variation achieved the highest precision within this training variation. The point clouds represent trees with predicted points indicating branches for pruning, with four different colors depicting the classification results: green for true positives (correctly predicted pruning points), gray for true negatives (correctly predicted non-pruning points), red for false positives (incorrectly predicted pruning points), and blue for false negatives (ground truth pruning points missed by the model). Among the test samples, three common scenarios emerged. First, some examples, like those shown in Figure 9, exhibit high precision, with the model accurately identifying pruning points.

Further, there are cases where the predicted cutting points are very close to the ground truth, as illustrated in Figure 10. In such cases introduction of the post-processing module which will use information about tree morphology and pruning rules could be beneficial. This task is a complex task which requires collaboration with agronomy experts and is for sure one of the topics for our future research. We hope that introduction of such post-processing module will be able to improve pruning predictions obtained by PointNet++ or any other used neural network or algorithm.

Some samples, such as those in Figure 11, demonstrate lower precision and recall, often due to inaccuracies or inconsistencies in the ground truth annotations itself and test models not resembling training set.

This visual assessment provides valuable insights into the model’s practical performance and highlights the importance of careful ground truth validation.

## 7. Discussion

The primary motivation of this study was to develop and evaluate an automated pipeline for predicting branches to be pruned in orchards. Since the entire process is interconnected, errors tend to accumulate from one stage to the next, highlighting the potential for targeted improvements at each step.

The pipeline begins with capturing RGB-D images in the orchard. An initial refinement could involve upgrading the imaging sensor, such as employing a Time-of-Flight (ToF) camera, which is less affected by ambient sunlight. This would enable longer imaging sessions throughout the day, not just during late evening hours. Furthermore, higher-quality sensors could produce less noisy images with greater resolution, resulting in more informative point clouds.

Some form of segmentation before the whole registration process would be beneficial, since it may help with the removal of the considered tree-unrelated structures, such as other trees and supporting trellis wires. Improving the subsequent tree reconstruction could involve automating and enhancing the process further. Currently, manual selection of images is required; automating this step by capturing images in a consistent pattern and using all available images for reconstruction would be beneficial. Additionally, implementing an automatic validation mechanism for the reconstruction process—based on metrics or consistency checks—could reduce reliance on manual visual inspection and subjective assessment. This is particularly challenging given the absence of a definitive ground truth. Validation could be performed after each registration phase and upon completion of the full scaffold model, ensuring the accuracy of each reconstruction stage.

The annotations are generated based on the differences between the pre- and post-pruning models, under the assumption that the only changes come from the pruned branches. In practice, however, some points may also be missing in the post-pruning model due to imperfect reconstruction. These missing points are not systematic-sometimes they occur on the trunk, other times on branch segments near the trunk—and the network does not register them as important, so it does not attempt to predict them. As a result, they appear as false negatives, which lowers recall thus creating a gap between precision and recall values. The annotation process can be refined by segmenting the tree into its constituent parts—branches, trunk, and other structures—so that only the branches are considered for annotation. This would improve model training by reducing possible confusion and emphasizing the network’s focus. Incorporating expert validation directly on the point clouds, accompanied by comparisons to annotations obtained automatically, would enhance the reliability of the training data.

In this dataset, the branches to be pruned are typically longer and grow vertically, a structure that the network demonstrates to recognize quite effectively. The results suggest that the model’s performance is not highly sensitive to voxel size, provided that the resolution is sufficient to preserve such prominent structures. Therefore, the choice of voxel size can be optimized as a trade-off between training duration and prediction accuracy. Further improvements could be achieved through post-processing of predictions—for example, filtering out points predicted on the trunk rather than the branches or enforcing pruning rules to eliminate anatomically inconsistent predictions. Additionally, fine-tuning the network’s parameters may enhance prediction quality. Expanding the dataset with more diverse and representative samples would also be advantageous, especially to capture edge cases and improve generalization.

Overall, the results demonstrate that the pipeline is meaningful and effective, achieving a precision of approximately 75%. Visual inspection of the predicted branches to be pruned indicates that the network produces plausible results, frequently identifying the correct branches with only a slight offset in the predicted cutting points.

Based on the average duration of the entire reconstruction process, as shown in Table 4—which includes preprocessing (grass removal) and point cloud registration—grass removal emerges as the primary bottleneck, as it can take several minutes per tree. In contrast, registration is performed in just a few seconds and is compatible with real-time applications. The annotation process, reported in Table 5 is relatively fast, taking approximately 4 s per tree; however, since it can be conducted offline, it is less critical for real-time performance. Regarding training times, as shown in Table 8, the process ranges from about 8 min to 1 h and 16 min. While this could be accelerated with higher-performance hardware, it is performed offline and thus does not impact the system’s operational efficiency. Testing durations are in the realm of real time, ranging from 2 to 22 s per test set—roughly one second per tree in the worst case—and can be further optimized with more powerful computing resources.

The entire pipeline presented in this work opens several avenues for future research. Improvements in sensor technology, automation of reconstruction validation, enhanced annotation methods, and advanced post-processing techniques all present opportunities to refine and extend this approach. Additionally, expanding the dataset to include more diverse tree structures, growth stages, and orchard conditions will further improve model robustness and generalization. Overall, the framework laid out here serves as a foundational step, encouraging ongoing development and innovation automated tree modeling and pruning prediction.

## 8. Conclusions

This study outlines a comprehensive pipeline for automated prediction of branches to be pruned, consisting of four stages. First, we created a dataset comprising RGB-D images, corresponding point clouds, reconstructed tree models generated via the proposed reconstruction process, and annotations of branches to be pruned. This dataset provides a valuable resource for future research on tree model reconstruction and pruning prediction, complete with benchmark results for comparative evaluation. Then, the reconstruction process is performed, which utilizes the TEASER++ algorithm, combined with automated preprocessing steps such as background and ground removal, to produce partial 3D models of the trees. After that, branches to be pruned are automatically labeled based on registering models before and after pruning, and identifying points that differ. This step enables the generation of training data for the prediction network. Finally, a deep learning network is trained to predict branches to be pruned, with performance assessed through quantitative metrics and qualitative visual inspections. The results demonstrate the approach’s viability, with the network frequently identifying relevant branches, although some inaccuracies remain.

For each step, potential limitations—such as sensor constraints, manual selection of images used in reconstruction, data-driven annotation, lack of post-processing, and dataset size and variability—have been identified, and improvements have been suggested. These considerations lay the groundwork for future investigations aimed at refining each component, enhancing accuracy, and expanding applicability.

## Figures and Tables

**Figure 1 sensors-25-05648-f001:**
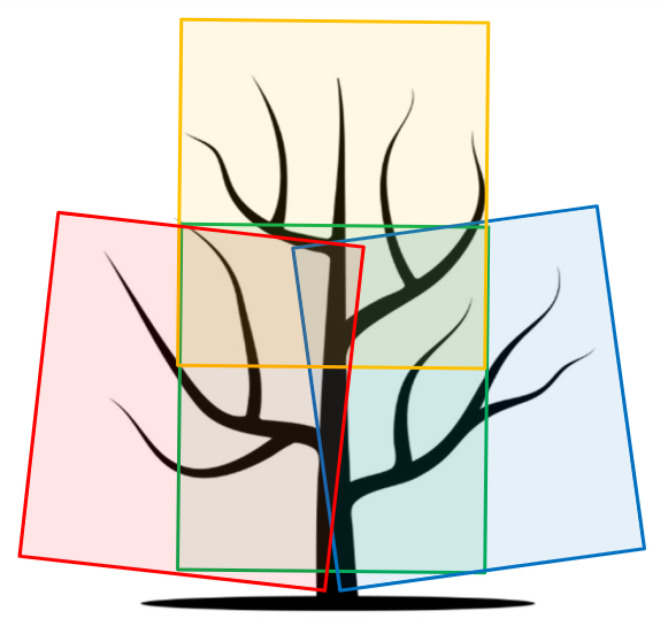
Example of images obtained from four viewpoint, denoted in red, yellow, green and blue.

**Figure 2 sensors-25-05648-f002:**
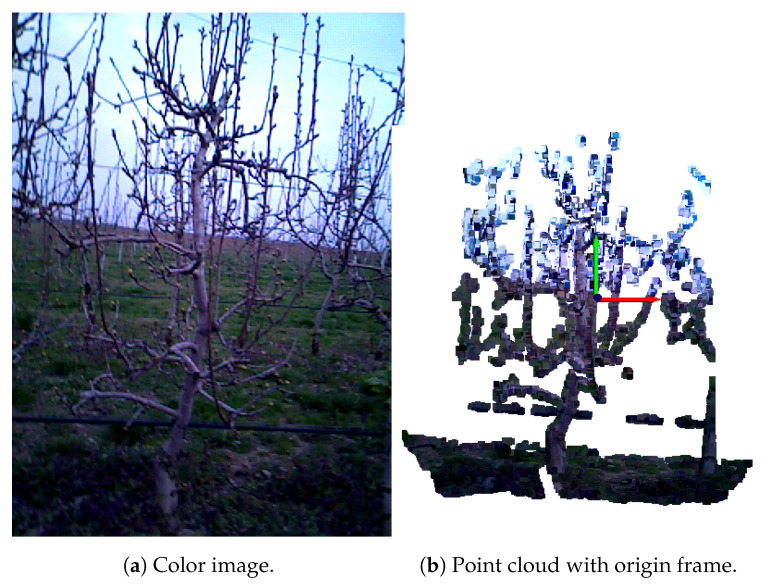
Example of a pear tree before pruning.

**Figure 3 sensors-25-05648-f003:**
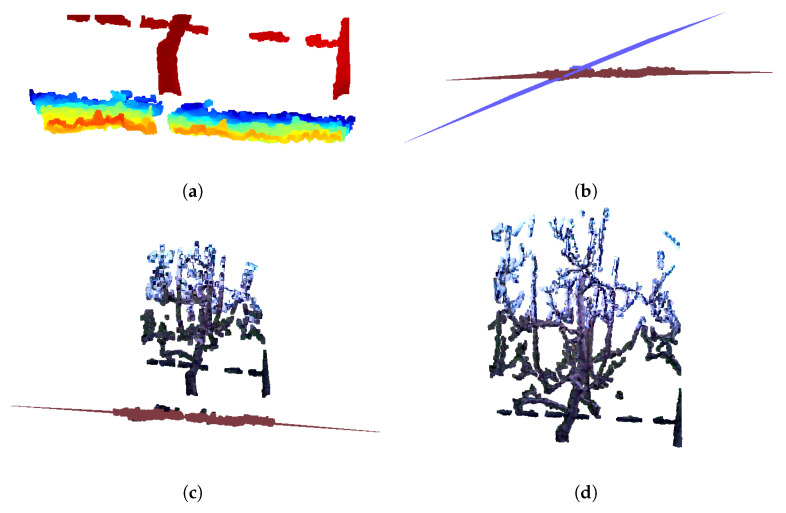
Preprocessing steps for pear tree before pruning. (**a**) Point cloud *B*, representing bottom part of the scene. Closer points are represented with warmer colors (red, yellow), farther points are represented with colder colors (green, blue). (**b**) Segmenting planes on *B* using segment_plane() Open3D method. (**c**) Segmented dominant plane representing grass on point cloud *P*. (**d**) Preprocessed tree point cloud.

**Figure 4 sensors-25-05648-f004:**
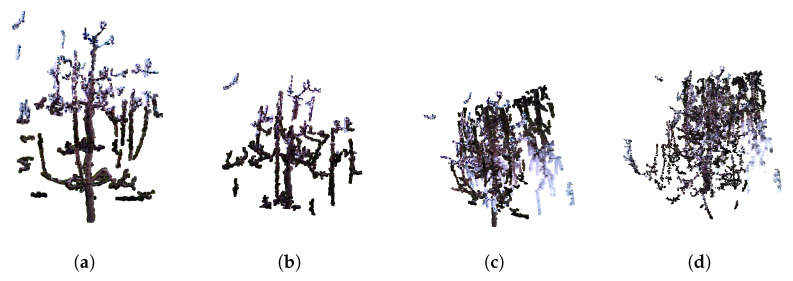
Illustration of the effect of point cloud selection on registration quality. (**a**) Registration using four manually selected point clouds, resulting in correct alignment. (**b**) Registration with only two selected point clouds, leading to incomplete reconstruction. (**c**) Registration with four randomly chosen point clouds, producing noticeable misalignments. (**d**) Registration using six point clouds, where redundant and noisy data caused registration failure.

**Figure 5 sensors-25-05648-f005:**
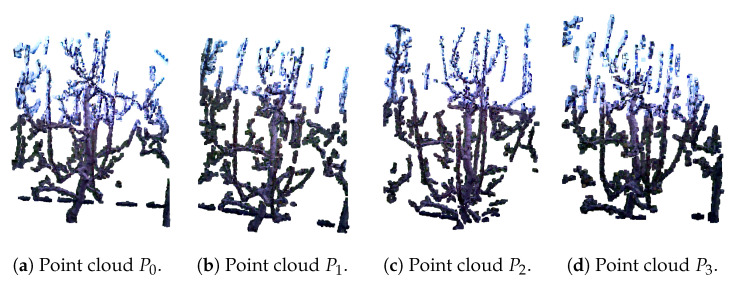
Preprocessed point clouds of a pear tree, obtained by the RGB-D images from 4 viewpoints, before pruning, prepared for reconstruction using Algorithm 2.

**Figure 6 sensors-25-05648-f006:**
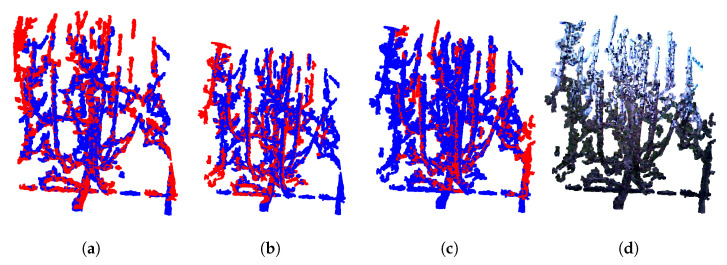
Example of the reconstruction process of the pear tree before pruning. (**a**) Registration of $P_{0}$ and $P_{1}$. (**b**) Registration of $P_{0,1}$ and $P_{2}$. (**c**) Registration of $P_{0,1,2}$ and $P_{3}$. (**d**) Colored registration of $P_{0,1,2,3}$ and $P_{4}$.

**Figure 7 sensors-25-05648-f007:**
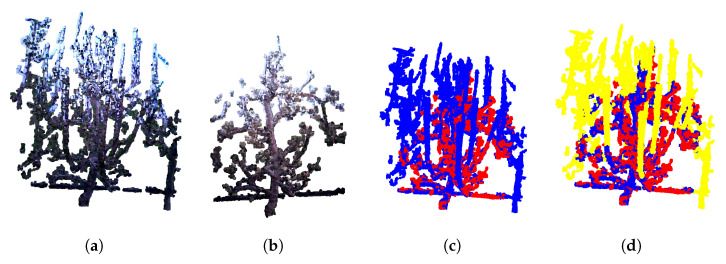
Example of the annotation process obtained by overlapping models pre- and post-pruning. (**a**) Reconstructed model pre-pruning. (**b**) Reconstructed model post-pruning. (**c**) Overlap of pre- and post-pruning models. (**d**) Annotated branches to be pruned (in yellow).

**Figure 8 sensors-25-05648-f008:**
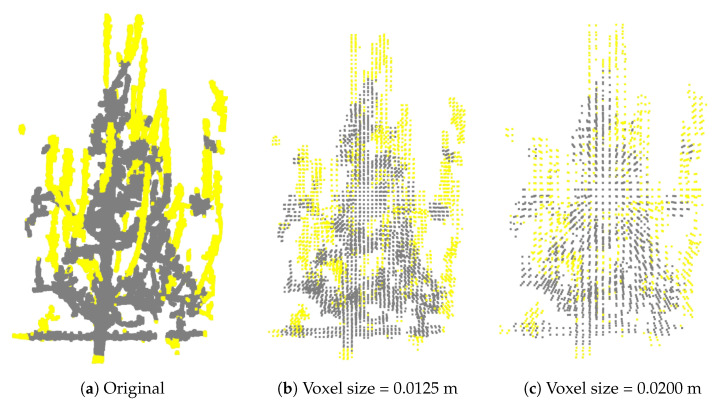
Comparison of the same model with different voxel sizes. Yellow points represent ground truth points belonging to the pruned branches.

**Figure 9 sensors-25-05648-f009:**
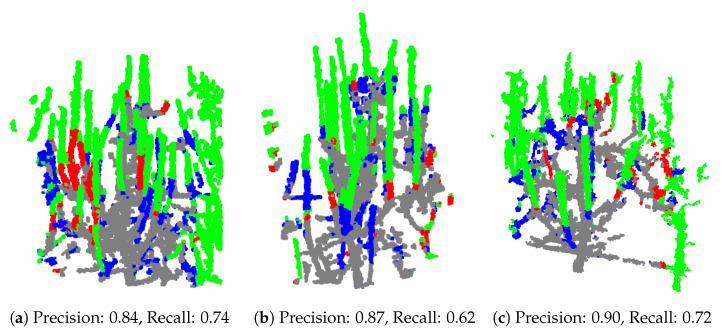
Examples of test models achieving high precision.

**Figure 10 sensors-25-05648-f010:**
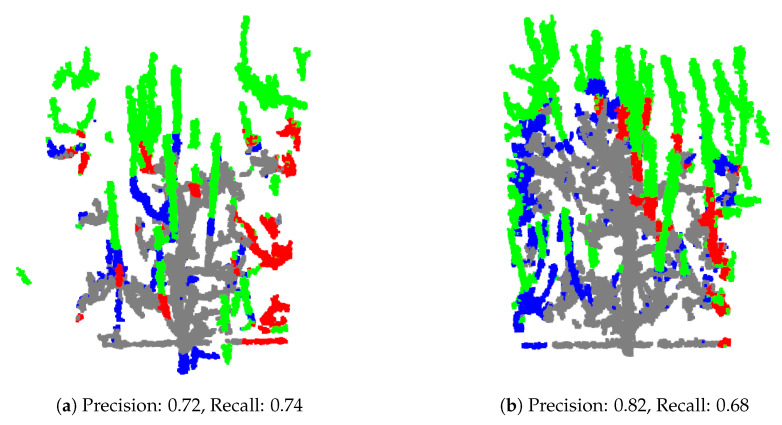
Examples of test models where the predicted branches to be pruned are correctly identified, but the ground truth cutting point is located slightly higher or lower along the branch, indicating a spatial discrepancy between the prediction and ground truth.

**Figure 11 sensors-25-05648-f011:**
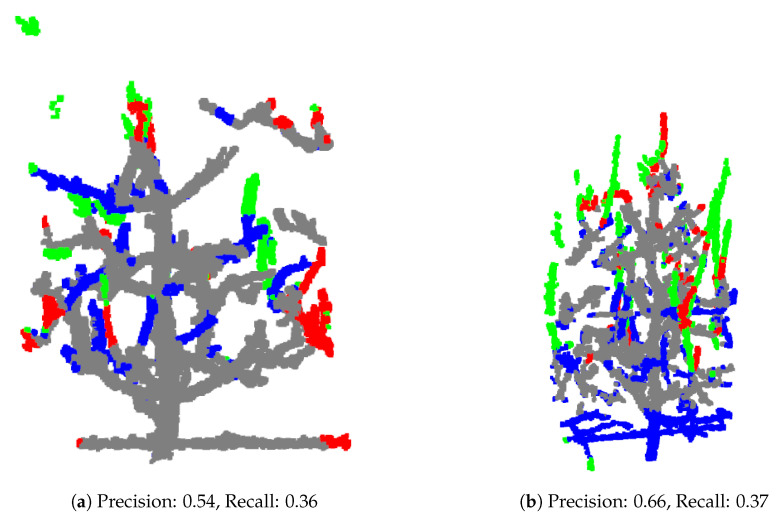
Test samples showcasing lower precision and recall, primarily due to inconsistencies between the training and test sets (**a**) or inaccuracies in the ground truth annotations (**b**), which affect the model’s prediction accuracy.

**Table 1 sensors-25-05648-t001:** Comparison of Datasets.

Source	Camera/Sensor	Dataset	Annotations
[2]	Environment for video game development	43,000 synthetic RGB-D images	Annotated trees, trunk cut location, tree diameter, and tilt
[3]	KinectV2 sensor	9 apple trees, 4 RGB-D images per tree	Annotated ground truth, main branch diameters, distances between neighboring branches
[4]	KinectV2 sensor	210 (training) + 90 (test) RGB-D images	Classified into background, trunk, and branches
[5]	KinectV2 sensor	270 RGB-D image sets	Manually annotated as ground truth
[6]	Intel RealSense L515, D435i, XREAL Light AR, Samsung Galaxy S20 FE 5G	11,000 RGB-D images	Two semantic masks: vines, branches, nodes, support wires, pruning regions, dried segments, buds
[7]	TLS sensors	Forest point clouds	Classified into six classes: ground, trunk, first-order branches, higher-order branches, leaves, and miscellaneous
[1]	ASUS Xtion PRO Live RGB-D camera	RGB-D images of 70 pear trees pre- and post-pruning, 52 reconstructed tree models pre- and post-pruning	Data driven per-point annotations of branches to be pruned
BRANCH_v2	ASUS Xtion PRO Live RGB-D camera	RGB-D images of 184 pear trees pre- and post-pruning, 101 reconstructed tree models pre- and post-pruning	Data driven per-point annotations of branches to be pruned

**Table 2 sensors-25-05648-t002:** Parameters of the TEASER++ robust registration solver used for generating point cloud registrations.

Parameter	Original Value	Adjusted Value
cbar	1	1
noise_bound	1	0.02
estimate_scaling	True	False
rotation_estimation_algorithm	GNC_TLS	GNC_TLS
rotation_gnc_factor	1.4	1.4
rotation_max_iterations	100	100
rotation_cost_threshold	$10^{-12}$	$10^{-12}$

**Table 3 sensors-25-05648-t003:** Overview of the reconstructed tree models in BRANCH_v2 dataset, showing the number of correctly reconstructed tree models based on multiple RGB-D images using TEASER++. In order to annotate points belonging to the pruned branches, both pre- and post-pruning models should be successfully reconstructed and correctly aligned.

	Pre-Pruning	Post-Pruning	Both Successfully Reconstructed Models	Annotated Models
Number of successfully registered point cloud sets	133	143	103	101
Percentage (%)	72.3	77.7	55.9	54.9

**Table 4 sensors-25-05648-t004:** Timing overview for reconstruction of tree models, which includes grass removal and registration of point clouds.

		Pre-Pruning			Post-Pruning		
		**Min**	**Max**	**Mean**	**Min**	**Max**	**Mean**
Grass removal time [s]		0.21	404.34	133.17	10.91	464.93	170.71
	$P_{0}-P_{1}$	0.11	3.72	0.65	0.15	1.25	0.49
Registration time [s]	$P_{0,1}-P_{2}$	0.17	3.87	0.67	0.15	1.41	0.54
	$P_{0,1,2}-P_{3}$	0.19	2.82	0.77	0.14	1.50	0.62

**Table 5 sensors-25-05648-t005:** Timing overview of the annotation process which includes registration between pre- and post-pruning models and annotation of points belonging to the branches to be pruned.

	Min	Max	Mean
Registration time [s]	0.58	12.18	3.28
Annotation time [s]	0.20	2.00	0.53

**Table 6 sensors-25-05648-t006:** Hyperparameters used during training of PointNet++.

Hyperparameter	Value
Batch size	[2, 4]
Learning rate	[0.0025, 0.005]
Optimizer	Adam
Weight decay	$10^{-4}$
Learning rate decay factor	0.5
Learning rate decay step	20 epochs
Number of epochs	100
Number of output classes	2 (‘1’—pruned, ‘0’—not pruned)
npoints (per point cloud)	[minimum, maximum]
Train/Val/Test ratio	0.7/0.1/0.2
Random seed	[10, 42]

**Table 7 sensors-25-05648-t007:** Number of Points in Dataset Variations.

Dataset	min_Points	max_Points
original	175,620	469,152
normalized_centered_model	175,620	469,152
voxelized_model_0.0025	62,673	151,925
voxelized_model_0.0050	19,677	53,536
voxelized_model_0.0075	9831	26,698
voxelized_model_0.0100	5987	16,256
voxelized_model_0.0125	4125	11,139
voxelized_model_0.0150	3032	8129
voxelized_model_0.0175	2402	6313
voxelized_model_0.0200	1943	5039

**Table 8 sensors-25-05648-t008:** Training and test information and results.

seed = 42										
**total number of models = 101, number of models in training/validation/test = 70/10/21**										
**batch_size = 2, learning_rate = 0.0025**										
**npoints = min_points**	**validation**					**test**				
	**accuracy**	**precision**	**recall**	**F1**	**training duration**	**accuracy**	**precision**	**recall**	**F1**	**test duration**
original	0.80	0.76	0.66	0.71	1 h 16 min 0 s	**0.80**	0.72	0.63	0.67	0 h 0 min 21 s
normalized_centered_model	0.83	0.71	0.60	0.64	1 h 16 min 50 s	0.79	0.69	0.63	0.66	0 h 0 min 22 s
voxelized_model_0.0025	0.80	0.77	0.61	0.67	0 h 28 min 53 s	0.78	**0.75**	0.57	0.65	0 h 0 min 9 s
voxelized_model_0.0050	0.79	0.74	0.66	0.69	0 h 12 min 55 s	0.76	0.71	0.60	0.65	0 h 0 min 4 s
voxelized_model_0.0075	0.78	0.75	0.66	0.70	0 h 10 min 45 s	0.76	0.71	0.66	0.68	0 h 0 min 2 s
voxelized_model_0.0100	0.78	0.75	0.67	0.71	0 h 9 min 51 s	0.75	0.69	0.67	0.68	0 h 0 min 2 s
voxelized_model_0.0125	0.78	0.76	0.65	0.70	0 h 7 min 38 s	0.76	0.72	0.69	**0.70**	0 h 0 min 2 s
voxelized_model_0.0150	0.77	0.73	0.68	0.70	0 h 6 min 54 s	0.74	0.70	0.70	0.70	0 h 0 min 2 s
voxelized_model_0.0175	0.76	0.74	0.66	0.69	0 h 6 min 47 s	0.74	0.71	0.68	0.69	0 h 0 min 1 s
voxelized_model_0.0200	0.76	0.76	0.63	0.68	0 h 8 min 0 s	0.75	0.71	**0.70**	**0.70**	0 h 0 min 2 s
**npoints = max_points**	**validation**					**test**				
	**accuracy**	**precision**	**recall**	**F1**	**training duration**	**accuracy**	**precision**	**recall**	**F1**	**test duration**
original	Could not be performed due to the insufficient GPU memory									
normalized_centered_models										
voxelized_model_0.0025	0.81	0.77	0.63	0.69	0 h 56 min 4 s	0.78	0.72	0.61	0.66	0 h 0 min 19 s
voxelized_model_0.0200	0.76	0.81	0.58	0.67	0 h 9 min 25 s	0.74	**0.77**	0.57	0.66	0 h 0 min 2 s
**batch_size = 4, learning_rate = 0.005**										
**npoints = min_points**	**validation**					**test**				
	**accuracy**	**precision**	**recall**	**F1**	**training duration**	**accuracy**	**precision**	**recall**	**F1**	**test duration**
original	Could not be performed due to the insufficient GPU memory									
normalized_centered_models										
voxelized_model_0.0025	0.81	0.74	0.68	0.71	0 h 26 min 3 s	0.77	0.67	0.67	0.67	0 h 0 min 19 s
voxelized_model_0.0050	0.79	0.80	0.62	0.68	0 h 10 min 55 s	0.78	0.75	0.61	0.67	0 h 0 min 4 s
voxelized_model_0.0075	0.78	0.74	0.67	0.70	0 h 8 min 13 s	0.75	0.68	0.65	0.67	0 h 0 min 3 s
voxelized_model_0.0100	0.78	0.75	0.68	0.71	0 h 6 min 6 s	0.76	0.73	0.65	0.69	0 h 0 min 2 s
voxelized_model_0.0125	0.77	0.76	0.66	0.70	0 h 6 min 31 s	0.75	0.72	0.65	0.68	0 h 0 min 2 s
voxelized_model_0.0150	0.76	0.75	0.65	0.68	0 h 5 min 24 s	0.74	0.70	0.70	0.70	0 h 0 min 2 s
voxelized_model_0.0175	0.76	0.79	0.60	0.68	0 h 5 min 24 s	0.74	0.71	0.68	0.69	0 h 0 min 2 s
voxelized_model_0.0200	0.76	0.75	0.69	0.71	0 h 5 min 2 s	0.73	0.68	0.69	0.68	0 h 0 min 2 s

## Data Availability

The original data presented in the study are openly available at https://puh.srce.hr/s/EoPqgASGerLapne (accessed on 7 September 2025).

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
