# Peer review of "Towards Robotic Pruning: Automated Annotation and Prediction of Branches for Pruning on Trees Reconstructed Using RGB-D Images"

_sensors, 2025, doi:10.3390/s25185648_

Round 1
Reviewer 1 Report
Comments and Suggestions for Authors
This paper pioneers a comprehensive pruning framework encompassing "acquisition-reconstruction-annotation-prediction" processes, with core contributions including: 1) An automatic annotation method for point cloud differentials before and after pruning; 2) TEASER++ adaptation for sparse tree registration; 3) BRANCH_v2 - the largest-scale pear tree pruning point cloud dataset to date. The 75% prediction accuracy validates its agricultural application potential, while the high-precision-first strategy aligns with production safety requirements.
The following are specific opinions:
- Insufficient justification for threshold τ:
Section 5.1 states the distance threshold τ=0.03m was selected empirically, but three critical gaps exist:- Lack of quantitative basis:No statistical distribution of registration error (e.g., RMSE, 95th percentile) was provided to demonstrate this value covers typical noise.
- Absence of sensitivity analysis:The impact of τ within the range [0.01, 0.05]m on annotation accuracy metrics (Precision/Recall) was not evaluated.
- Agronomic suitability not argued:The biological rationale for the threshold value was not explained by linking it to horticultural standards (e.g., safety length standards for residual branch stubs).
- Reconstruction error not quantified:
The success of TEASER++ registration (Section 3.2) was determined solely via subjective visual assessment. Objective accuracy metrics (e.g., RMSE after point cloud registration, relative pose error) were not reported. - "Real-time" claim lacks support:
The claim of being "suitable for real-time robotic systems" (Abstract and Section 6) is not substantiated by data on the timing of critical steps (e.g., single-tree reconstruction time, model inference latency). - Weak data augmentation strategy:
Given the small dataset size (N=101), only basic preprocessing was applied. Robust augmentation methods to counter environmental interference (e.g., lighting perturbation, occlusion simulation) were not introduced. - Lack of developed post-processing mechanism:
Systematic spatial shifts exist in the prediction results (Figure 9), yet no post-processing module leveraging prior knowledge of tree morphology (e.g., the rule that "pruning points are located on lateral branches") was designed. - Insufficient reproducibility safeguards:
The dataset and code are stated to be "available upon request". To enhance reproducibility, it is strongly recommended to openly publish the dataset and code.
Reviewer 2 Report
Comments and Suggestions for Authors
Please find the attached file.

Round 2
Reviewer 2 Report
Comments and Suggestions for Authors
I have carefully reviewed the revised version of the manuscript, and I am satisfied with the improvements and modifications made by the authors. The revisions effectively address the concerns raised in the previous review, and the explanations, methodological clarifications, and additional details provided enhance the overall quality of the paper. I have no further comments or suggestions for improvement, and I believe the paper is now suitable for publication in its current form.